

# Distribution of *Porphyromonas gingivalis* *fimA* and *mfa1* fimbrial genotypes in subgingival plaques

Keiji Nagano[1], Yoshiaki Hasegawa[1], Yura Iijima[1], Takeshi Kikuchi[2] and Akio Mitani[2]

[1] Department of Microbiology, School of Dentistry, Aichi Gakuin University, Nagoya, Japan
[2] Department of Periodontology, School of Dentistry, Aichi Gakuin University, Nagoya, Japan

Corresponding author
Keiji Nagano, nagano@dpc.agu.ac.jp

## ABSTRACT

**Background:** Strains of periodontal disease-associated bacterium *Porphyromonas gingivalis* have different pathogenicity, which can be attributed to clonal genetic diversity. *P. gingivalis* typically expresses two types of fimbriae, FimA and Mfa1, which comprise six (I, Ib, II, III, IV, and V) and two ($mfa^{53}$ and $mfa^{70}$) genotypes, respectively. This study was conducted to investigate the distribution of the two fimbrial genotypes of *P. gingivalis* in clinical specimens.

**Methods:** Subgingival plaques were collected from 100 participants during periodontal maintenance therapy and examined for *P. gingivalis* fimbrial genotypes by direct polymerase chain reaction and/or DNA sequencing. We also analyzed the relationship between fimbrial genotypes and clinical parameters of periodontitis recorded at the first medical examination.

**Results:** Both fimbrial types could be detected in 63 out of 100 samples; among them, *fimA* genotype II was found in 33 samples (52.4%), in which the $mfa^{70}$ genotype was 1.75 times more prevalent than $mfa^{53}$. The total detection rate of *fimA* genotypes I and Ib was 38.1%; in these samples, the two *mfa1* genotypes were observed at a comparable frequency. In two samples positive for *fimA* III (3.2%), only $mfa^{53}$ was detected, whereas in four samples positive for *fimA* IV (6.3%), the two *mfa1* genotypes were equally represented, and none of *fimA* V-positive samples defined the *mfa1* genotype. No associations were found between clinical parameters and fimbrial subtype combinations.

**Discussion:** Both *P. gingivalis* fimbrial types were detected at various ratios in subgingival plaques, and a tendency for *fimA* and *mfa1* genotype combinations was observed. However, there was no association between *P. gingivalis* fimbrial genotypes and periodontitis severity.

## INTRODUCTION

Periodontal diseases are developed because of colonization of the subgingival area by multiple bacterial species (*Page & Kornman, 1997*). *Socransky et al. (1998)* have determined that three bacterial species, *Porphyromonas gingivalis*, *Tannerella forsythia*, and *Treponema denticola*, are mostly responsible for the development and advancement

of periodontitis, and proposed to call them the "red complex." Among these species, *P. gingivalis*, a Gram-negative anaerobic bacterium forming characteristic black-pigmented colonies on blood agar, has been extensively studied for its pathogenicity (*Gibbons & Macdonald, 1960*; *Macdonald & Gibbons, 1962*; *Macdonald, Gibbons & Socransky, 1960*), and accumulated evidence indicates its critical role in periodontitis (*Lamont & Jenkinson, 1998*; *Socransky & Haffajee, 2002*). Furthermore, although the proportion of *P. gingivalis* in the periodontal biofilm is low, it could lead to dysbiosis at the periodontal site, which prompted *Hajishengallis et al. (2011)* to call *P. gingivalis* a keystone pathogen. Still, it is well known that *P. gingivalis* can also be detected in healthy people (*Amano et al., 2004*; *Griffen et al., 1998*; *Haffajee et al., 1998*; *Teanpaisan et al., 1996*; *Ximenez-Fyvie, Haffajee & Socransky, 2000*), suggesting that its presence may not necessarily cause periodontitis. This discrepancy is suggested to be attributed to heterogenic virulence of *P. gingivalis* (*Griffen et al., 1999*; *Igboin, Griffen & Leys, 2009*; *Tribble, Kerr & Wang, 2013*), which shows a high degree of genetic clonal diversity (*Enersen, 2011*).

Although *P. gingivalis* expresses a number of potential virulence factors (*How, Song & Chan, 2016*), fimbriae, filamentous proteinaceous appendages on the bacterial surface, are one of the most important because they play a pivotal role in *P. gingivalis* colonization through association with other bacteria and host tissues (*Hospenthal, Costa & Waksman, 2017*; *Lamont & Jenkinson, 2000*). *P. gingivalis* generally expresses two distinct types of fimbriae: FimA and Mfa1 (*Yoshimura et al., 2009*). FimA fimbriae are primarily composed of polymers of the FimA protein encoded by the *fimA* gene (*Dickinson et al., 1988*; *Yoshimura et al., 1984*), whereas Mfa1 fimbriae are mostly composed of the Mfa1 protein encoded by the *mfa1* gene (*Hamada et al., 1996*). In addition, several minor accessory components are incorporated into the respective fimbriae (*Hasegawa et al., 2013*; *Nishiyama et al., 2007*).

Based on *fimA* sequence variability, the gene is classified into six genotypes (I, Ib, II, III, IV, and V) (*Amano et al., 2004*; *Nakagawa et al., 2002b*), and the encoded proteins exhibited distinct antigenicity, with the exception of subtypes I and Ib (*Nagano et al., 2013*; *Nakagawa et al., 2002b*). Several studies indicate that strains with *fimA* genotype II are the most prevalent in patients with periodontitis, whereas those with genotype I are predominantly detected in healthy individuals (*Amano et al., 2004*; *Enersen, Nakano & Amano, 2013*; *Kuboniwa, Inaba & Amano, 2010*; *Missailidis et al., 2004*; *Miura et al., 2005*), indicating that the genotype-II *P. gingivalis* may have higher pathogenicity compared with genotype-I bacteria. Furthermore, genotype-II strains showed higher adhesion and invasion ability in human epithelial cells (*Nakagawa et al., 2002a*) and in a mouse abscess model (*Nakano et al., 2004*). However, other reports indicated that *fimA* genotypes were not associated with adhesion to and invasion of host cells (*Inaba et al., 2008*; *Umeda et al., 2006*); moreover, there are studies showing that genotype-II strains had rather low rates of adhesion to and invasion of epithelial cells (*Eick et al., 2002*) and induced significantly less alveolar bone resorption in mice compared to genotype-I strains (*Wang et al., 2009*). Collectively, these data indicate that *P. gingivalis* pathogenicity cannot be defined based on the *fimA* genotype.

In contrast to *fimA*, there are few clinical data regarding *mfa1* genotypes. Recently, we found that the *mfa1* gene had at least two variants (*Nagano et al., 2015*), encoding proteins with molecular weights about 70 kDa (67 (*Arai, Hamada & Umemoto, 2000*; *Hamada et al., 1996*) or 75 kDa (*Park et al., 2005*)) and 53 kDa (*Arai, Hamada & Umemoto, 2000*; *Nagano et al., 2015*), hereafter called Mfa$^{70}$ and Mfa$^{53}$ (*mfa$^{70}$* and *mfa$^{53}$*, respectively, for the genes). In this study, we investigated the distribution of *P. gingivalis mfa1* as well as *fimA* genotypes in clinical specimens.

## MATERIALS AND METHODS

### Participants

A total of 100 patients, who visited Aichi Gakuin University Dental Hospital at Nagoya, Japan, for periodontal treatment from September 2016 to March 2017, participated in this study. The study was approved by the institutional review board (Aichi Gakuin University, School of Dentistry, Ethics Committee, approval numbers 460 and 478), and written informed consent was obtained from all participants.

### Clinical oral examination, and consolidation and maintenance treatments

Among the 100 participants, 81 could be examined for clinicopathological parameters of periodontitis at the first visit. Clinical oral examination was performed according to the guidelines published by *The Japanese Society of Periodontology (2015)*. Probing pocket depth (PD) and bleeding on probing (BOP) analyzed in six sites per tooth (buccal-mesial, mid-buccal, buccal-distal, lingual-mesial, mid-lingual, and lingual-distal) for all remaining teeth. The PD and BOP values were utilized to calculate periodontal inflamed surface area (PISA) and periodontal epithelial surface area (PESA), which reflect the surface area of bleeding epithelium and total pocket epithelium (in mm$^2$), respectively, using a free spreadsheet (downloaded from www.parsprototo.info) (*Nesse et al., 2008*, *2009*). Consolidation and maintenance treatments mainly consisted of professional scaling and cleaning. Patients visited the hospital for consultation every 1–6 months.

### Collection of subgingival plaques

Subgingival plaque samples were collected by a sterile hand scaler and transferred in either one ml of sterile reduced transfer fluid (RTF) consisting of 0.01% dithiothreitol in PBS, pH 7.4 or one ml of distilled water. The samples were immediately placed at 4 °C and analyzed within 4 h.

### Isolation and identification of black-pigmented bacteria

The samples collected in RTF were thoroughly suspended, serially diluted, and aliquots spread on blood agar consisting of Brucella HK agar (Kyokuto Pharmaceutical Industrial Co., Ltd, Tokyo, Japan), 5% laked rabbit blood, and 100 µg/ml kanamycin, as anaerobic bacteria, including *P. gingivalis*, are typically kanamycin resistant (*Jousimies-Somer et al., 2002*). Plates were cultured at 37 °C under anaerobic conditions

for a week, and the emerged black-pigmented colonies were streaked on fresh plates to ensure isolation of single clones, which were then subjected to species identification. For this, genomic DNA was purified using Wizard Genomic DNA Purification Kit (Promega Corporation, Madison, WI, USA) and analyzed for 16S rRNA-encoding genes by polymerase chain reaction (PCR) using primers (5′-GAAGAGTTTGATCMTGG CTCAGATTG-3′ and 5′-TACGGYTACCTTGTTACGACTTCAC-3′) slightly modified from the universal primers 27F and 1492R (*Frank et al., 2008*). PCR products were subjected to DNA sequencing by a dye-terminator method and sequencing reads were analyzed by the BLAST search (https://blast.ncbi.nlm.nih.gov/). Bacterial species were identified if samples showed the lowest expectation ($E$) value (i.e., the highest probability) in the list of BLAST results. Most of $E$ values were 0, whereas the highest was $3 \times 10^{-66}$, i.e., were sufficiently low to identify bacterial species.

### Genotyping of *fimA* and *mfa1*

*FimA* genotypes were determined by PCR, sequencing, and BLAST analysis. Plaque samples in RTF or water were directly used as PCR templates. Primers for PCR (5′-AGCTTGTAACAAAGACAACGAGGCAG-3′ and 5′-GAGAATGAATACGGGGAG TGGAGCG-3′) were designed for common *fimA* regions based on *fimA* sequencing data for 84 *P. gingivalis* strains (*Nagano et al., 2013*). PCR-amplified fragments of a predicted size (around 1.2 kb) were sequenced by the dye-terminator method and the *fimA* genotype was determined by BLAST analysis.

  *Mfa1* genotypes were determined by PCR using two primer sets (5′-GAGCATTG CTCTCATTGGGCTTTG-3′ and 5′-CATCAGAAAAGGCAGCGTAAGCTG-3′, and 5′-GAGCATTGCTCTCATTGGGCTTTG-3′ and 5′-TTAGGTATTGGCGACG TTCTCCTTG-3′), which yielded $mfa^{53}$ and $mfa^{70}$ fragments of 410 and 830 bp, respectively.

### Statistical analysis

The data were expressed as the mean ± SEM. Differences between groups were analyzed by the nonparametric Kruskal–Wallis $H$ test, and were considered statistically significant at $P < 0.01$. The Chi square test is used to determine if there is a relationship in the genotype distribution ($P < 0.01$).

## RESULTS

### Isolation of *P. gingivalis*

The first 73 dental plaque samples were collected in RTF to isolate *P. gingivalis* by a culture method. Although black-pigmented colonies were obtained from the majority of samples, 16S rDNA sequencing analysis showed that they were mostly formed by *Prevotella* species, and the isolation rate of *P. gingivalis* was only 5.5%. Therefore, we decided to examine fimbrial genotypes by direct PCR; in addition, the collection solution was changed to water, did not affect experimental results, but slightly improved detectability.

**Table 1 Genotype distribution of fimbriae-encoding genes *fimA* and *mfa1*.**

| *fimA* | *mfa1* | | | Total (%) |
|---|---|---|---|---|
| | $mfa^{53}$ | $mfa^{70}$ | Undetermined | |
| I | 4 | 3 | (1) | 7 (11.1) |
| Ib | 8 | 9 | (3) | 17 (27.0) |
| I + Ib | 12 | 12 | (4) | 24 (38.1) |
| II | 12 | 21 | (6) | 33 (52.4) |
| III | 2 | 0 | (0) | 2 (3.2) |
| IV | 2 | 2 | (1) | 4 (6.3) |
| V | 0 | 0 | (1) | 0 (0) |
| Undetermined | (0) | (3) | – | – |
| Total (%) | 28 (44.4) | 35 (55.6) | – | 63 (100) |

Notes:
Genes encoding both fimbrial types were determined in 63 of 100 samples. Samples marked "undetermined" were not included in the total numbers.

**Table 2 Clinicopathological parameters of 81 study participants.**

| *fimA* | *mfa1* | Females (*n*) | Males (*n*) | Age (years) | PD (mm) | | BOP (%) | PISA ($mm^2$) | PESA ($mm^2$) |
|---|---|---|---|---|---|---|---|---|---|
| | | | | | Mean | Max | | | |
| Untyped | | 22 | 9 | 60.5 ± 2.5 | 3.22 ± 0.16 | 8.94 ± 0.51 | 44.2 ± 6.7 | 703 ± 135 | 1,640 ± 114 |
| I + Ib | 53 | 5 | 3 | 51.5 ± 4.1 | 2.91 ± 0.16 | 7.88 ± 0.61 | 38.9 ± 10.7 | 462 ± 136 | 1,436 ± 168 |
| | 70 | 9 | 2 | 59.1 ± 3.8 | 3.73 ± 0.34 | 9.40 ± 0.85 | 73.1 ± 13.0 | 1,016 ± 244 | 1,951 ± 191 |
| II | 53 | 6 | 4 | 62.1 ± 3.1 | 3.21 ± 0.37 | 8.60 ± 0.92 | 50.7 ± 18.8 | 868 ± 407 | 1,752 ± 296 |
| | 70 | 8 | 10 | 59.6 ± 2.0 | 3.50 ± 0.17 | 9.67 ± 0.48 | 49.3 ± 9.2 | 889 ± 155 | 1,839 ± 156 |
| III | 53 | 1 | 1 | 72.0 | 2.70 | 8.50 | 18.0 | 342 | 1,418 |
| | 70 | 0 | 0 | – | – | – | – | – | – |
| IV | 53 | 1 | 0 | 18 | 1.80 | 5.00 | 27.0 | 193 | 918 |
| | 70 | 2 | 0 | 68 | 2.60 | 6.00 | 52.0 | 519 | 1,395 |
| V | 53 | 0 | 0 | – | – | – | – | – | – |
| | 70 | 0 | 0 | – | – | – | – | – | – |

## Distribution of *fimA* and *mfa1* genotypes

The distribution of fimbrial *fimA* and *mfa1* genotypes is summarized in Table 1. Among the 100 samples, both fimbrial types were detected in 63 and a single type in 15 samples, whereas 22 had no fimbrial genes. *FimA* genotype II was the most prevalent and detected in 33 of the 63 samples positive for both fimbrial genes (52.4%), followed by genotypes Ib and I detected in 27.0% and 11.1% samples, respectively, whereas the frequency of the other *fimA* genotypes was low. Although there was no statistically significant difference in combination of the *fimA* and *mfa1* genotypes, the following tendency was observed. The $mfa^{53}$ and $mfa^{70}$ genotypes were detected at comparable frequencies (44.4% and 55.6%, respectively) and each of them showed almost the same frequency in samples positive for *fimA* genotypes I, Ib, and IV. However, the prevalence of $mfa^{70}$ was 1.75 times higher than that of $mfa^{53}$ in genotype-II positive samples,

whereas only $mfa^{53}$ was detected in the two genotype III-positive samples, and no *mfa1* genes were found in genotype V-positive samples.

## Relationship between clinical parameters and fimbrial genotypes

We also examined the association of the fimbrial genotypes with clinical characteristics of periodontitis (maximal and mean PD values, and BOP, PISA, and PESA values) (Table 2). However, no statistically significant differences in periodontitis severity were observed depending on the fimbrial genotypes.

## DISCUSSION

In this study, we first attempted to isolate *P. gingivalis* from dental plaque samples by a culture method, because we thought that analysis of chromosomal DNA purified from isolated bacterial clones by PCR would provide unequivocal fimbrial genotyping results. However, *P. gingivalis* was rarely isolated by the culture method. On the other hand, direct PCR detected either *fimA* or *mfa1* in 78% samples, indicating that *P. gingivalis* was present with high frequency in patients receiving periodontal maintenance therapy, although its proportion among dental plaque bacteria was low.

*FimA* genotypes have been determined by PCR using genotype-specific primers (*Amano et al., 2004*; *Nakagawa et al., 2002b*); in addition, restriction enzyme digestion is used to discriminate genotypes I and Ib (*Nakagawa et al., 2002b*), which, however, may not be necessary for the analysis of the entire *fimA* gene, because genotypes I and Ib cannot be clearly discriminated (Fig. S1 and *Nagano et al., 2013*). Furthermore, immunological analysis did not detect any differences in antigenicity between FimA I and Ib fimbriae (*Nagano et al., 2013*; *Nakagawa et al., 2002b*). Therefore, we do not discuss differences between genotypes I and Ib here. In contrast, genotype II (and possibly IV) could be further divided into two or more groups (Fig. S1 and *Nagano et al., 2013*). Regarding *mfa1*, two genotypes are currently known: $mfa^{53}$ and $mfa^{70}$. However, in 12% of *fimA*-positive specimens, *mfa1* was not detected, suggesting that existence of additional *mfa1* genotypes. Therefore, reclassification of *fimA* and *mfa1* genotypes would be needed in the future.

In this study, we observed that in samples positive for *fimA* genotype II, $mfa^{70}$ genotype was detected 1.75 times more frequently compared to $mfa^{53}$, and in the previous study, where we analyzed 84 *P. gingivalis* strains stocked in our laboratory, the frequency of $mfa^{70}$ detection among *fimA* II strains was 3.6 times higher than that of $mfa^{53}$ (*Nagano et al., 2015*). These findings indicate that $mfa^{70}$ is the major *mfa1* genotype in *P. gingivalis* strains positive for *fimA* II. On the other hand, in this study, the two *mfa1* genotypes had almost the same detection rate in samples positive for *fimA* I (including I and Ib), whereas our previous results indicate that $mfa^{70}$ detection frequency was 2.3 times higher compared to that of $mfa^{53}$ in *fimA*-I strains (*Nagano et al., 2015*). Among *fimA* IV-positive samples, the detection rate of each *mfa1* genotype was the same in this study, and in our previous study, $mfa^{53}$ and $mfa^{70}$ genotypes were detected in four and two samples, respectively (*Nagano et al., 2015*). Taken together, these results suggest that strains with *fimA* genotypes I and IV tend to have either the same frequency of *mfa1*

genotypes or slightly higher prevalence of $mfa^{70}$. Although we found only two genotype III-positive samples in this study, both had $mfa^{53}$, which was consistent with our earlier findings that 12 out of 13 genotype-III strains carried $mfa^{53}$ (*Nagano et al., 2015*). In this study, *mfa1* was not detected in genotype V-positive samples, which, however, were all found $mfa^{53}$-positive in our previous study (*Nagano et al., 2015*). These results indicate that genotype-III and -V strains almost exclusively carry $mfa^{53}$. Thus, there is a tendency for correlation between the two fimbrial types in *P. gingivalis*: *fimA* II strains preferably carry $mfa^{70}$, whereas *fimA* I/IV strains may have both *mfa1* genotypes in equal proportions, and *fimA* III/V strains mostly carry $mfa^{53}$. However, the reason for such correlations is unknown because there is a wide distance between the two genetic loci. There are polymorphisms in other *P. gingivalis* genes (*Enersen, 2011*). Thus, the *ragA* gene, which encodes a major outer membrane protein and is located downstream of the *mfa1* gene, exhibits four genetic variants (*Hall et al., 2005*; *Liu et al., 2013*); besides, genetic variability has also been reported for capsular antigens (*Laine, Appelmelk & Van Winkelhoff, 1996*; *Laine, Appelmelk & Van Winkelhoff, 1997*). In future studies, it will be interesting to find out whether these genetic polymorphisms are correlated with those in the *fimA* and *mfa1* genes.

We did not observe statistically significant associations between clinical parameters of periodontitis and the distribution of fimbrial genotypes. However, there was a time lapse between periodontal examination and sample collection, and it was possible that *P. gingivalis* clones were replaced during that interval; still, the chances for such clonal change may be low because it is reported that *P. gingivalis* showed high clonal stability (*Valenza et al., 2009*; *Van Winkelhoff, Rijnsburger & Van Der Velden, 2008*). In addition, we would like to note that most similar studies have the same problems which are inherent to this type of clinical research, because generally the treatment for chronic periodontitis takes a long time. Therefore, it is necessary to develop an appropriate study design for examining the relationship between bacterial genotypes and clinical symptoms of periodontitis.

## CONCLUSIONS

There was a tendency in the distribution of fimbrial genotypes *fimA* (I–V) and *mfa1* ($mfa^{53}$ and $mfa^{70}$) among patients with periodontitis. However, we did not observe any associations between fimbrial genotypes and the severity of the disease.

## ACKNOWLEDGEMENTS

We sincerely thank dental doctors in Aichi Gakuin University Dental Hospital for their cooperation in sample collection.

### Funding

This study was supported by the Aichi Health Promotion Foundation and JSPS KAKENHI Grant Number 16K11465 (KN). There was no additional external funding received for this

study. The funders had no role in study design, data collection and analysis, decision to publish, or preparation of the manuscript.

## Grant Disclosures
The following grant information was disclosed by the authors:
Aichi Health Promotion Foundation and JSPS KAKENHI: 16K11465.

## Competing Interests
The authors declare that they have no competing interests.

## Author Contributions
- Keiji Nagano conceived and designed the experiments, performed the experiments, analyzed the data, contributed reagents/materials/analysis tools, prepared figures and/or tables, authored or reviewed drafts of the paper, approved the final draft.
- Yoshiaki Hasegawa conceived and designed the experiments, performed the experiments, analyzed the data, contributed reagents/materials/analysis tools, authored or reviewed drafts of the paper, approved the final draft.
- Yura Iijima performed the experiments, contributed reagents/materials/analysis tools, authored or reviewed drafts of the paper, approved the final draft.
- Takeshi Kikuchi conceived and designed the experiments, performed the experiments, analyzed the data, authored or reviewed drafts of the paper, approved the final draft.
- Akio Mitani conceived and designed the experiments, performed the experiments, analyzed the data, authored or reviewed drafts of the paper, approved the final draft.

## Human Ethics
The following information was supplied relating to ethical approvals (i.e., approving body and any reference numbers):

The study was approved by Aichi Gakuin University, School of Dentistry, Ethics Committee, approval numbers 460 and 478.

## Data Availability
The raw data are provided in a Supplemental File.

## Supplemental Information
Supplemental information for this article can be found online at http://dx.doi.org/10.7717/peerj.5581#supplemental-information.

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
