# Peer review of "Distribution of *Porphyromonas gingivalis* *fimA* and *mfa1* fimbrial genotypes in subgingival plaques"

_PeerJ, doi:10.7717/peerj.5581_

## Round 0.1 · original submission · Minor Revisions

Dear Dr. Nagano and colleagues:

Thanks for submitting your manuscript to PeerJ. I apologize for the lengthy review process, as we had trouble finding reviewers. However, I have received two independent reviews of your work, and as you will see, the reviewers raised some concerns about the research. Thus, I extend to you the opportunity to revise your work.

Please consider the comments about experimental design, missing statistical analyses, explanation of protocols and approaches, data availability, and minor improvements to overall presentation.

I am recommending that you revise your manuscript accordingly, taking into account all of the issues raised by the reviewers. I do believe that your manuscript will be ready for publication once these issues are addressed.

Good luck with your revision,

-joe

·

Basic reporting

This is a well described manuscript from a corresponding author who has been doing pioneering work in the area of P. gingivalis fimbrial protein biochemistry and genetics. The findings of the study challenge some of the previous reports indicating that certain types of fimA genotypes (such as type II) are strongly associated with severity of periodontitis. The study is significant and its scientific merit is high.

Experimental design

Specific comments:
1. You performed sequencing of 16S rDNA amplicons followed by BLAST search. Are the results of this analysis included in the supplemental File 2? (please mention this in the text).
2. Thanks for providing the raw data and calculations in the Excel files. However, it is difficult to grasp what each column/row represents and what is being done to the data. It would be nice if you could provide clear identifiers and some description in the figure legends as to what each File/sheet presents (currently there are no figure legends).

Validity of the findings

Overall, the study was performed with high rigor, data are presented well and proper conclusions are drawn.

Additional comments

This is a well described manuscript from a corresponding author who has been doing pioneering work in the area of P. gingivalis fimbrial protein biochemistry and genetics. The findings of the study challenge some of the previous reports indicating that certain types of fimA genotypes (such as type II) are strongly associated with severity of periodontitis. The study is significant and its scientific merit is high.

Reviewer 2 ·

Basic reporting

The manuscript by Nagano et al examined association between P. gingivalis fimbrial genotypes and periodontitis severity. Experiments indicated that there was no association between fimbrial genotypes and periodontitis severity. Major concern is the novelty of this study, since several similar studies have been reported.

Experimental design

1. Ln 151-152, please indicate why dental plaque samples were resuspended in either RTF or water.
2. Ln 161, need details on DNA purification.
3. Ln 172-173, “Plaque samples in RTF or water were directly used as PCR templates”. How did you collect DNA?
4. Ln204-206, “Among the 100 samples, both fimbrial types were detected in 63 and a single type in 15 samples, whereas 22 had no fimbrial genes. Why not use Pg 16s to determine Pg level. Is no fimbrial gene means no Pg?

Validity of the findings

Table 1 did not include statistical analysis.

---

## Round 0.2 · accepted · Accept

Dear Dr. Nagano and colleagues:

Thanks for revising your manuscript based on the minor concerns raised by the reviewers. I now believe that your manuscript is suitable for publication. Congratulations! I look forward to seeing this work in print, and I anticipate it being an important resource for the Porphyromonas gingivalis pathogenesis community. Thanks again for choosing PeerJ to publish such important work.

Best,

-joe

#